# Human Protoparvovirus DNA and IgG in Children and Adults with and without Respiratory or Gastrointestinal Infections

**DOI:** 10.3390/v13030483

**Published:** 2021-03-15

**Authors:** Ushanandini Mohanraj, Maija Jokinen, Rajita Rayamajhi Thapa, Minna Paloniemi, Timo Vesikari, Maija Lappalainen, Eveliina Tarkka, Zaiga Nora-Krūkle, Anda Vilmane, Kim Vettenranta, Charles Mangani, Sami Oikarinen, Yue-Mei Fan, Per Ashorn, Elina Väisänen, Maria Söderlund-Venermo

**Affiliations:** 1Department of Virology, University of Helsinki, 00290 Helsinki, Finland; maija.jokinen@ieu.uzh.ch (M.J.); rajita.rayamajhithapa@helsinki.fi (R.R.T.); elina.vaisanen@thl.fi (E.V.); maria.soderlund-venermo@helsinki.fi (M.S.-V.); 2Faculty of Medicine and Health Technology, Tampere University, 33100 Tampere, Finland; minna.paloniemi@fimnet.fi (M.P.); sami.oikarinen@tuni.fi (S.O.); yuemei.fan@tuni.fi (Y.-M.F.); per.ashorn@tuni.fi (P.A.); 3Nordic Research Network Oy, 33700 Tampere, Finland; timo.vesikari@nrnetwork.fi; 4Helsinki University Hospital Laboratory (HUSLAB), 00290 Helsinki, Finland; maija.lappalainen@hus.fi (M.L.); eveliina.tarkka@hus.fi (E.T.); 5Institute of Microbiology and Virology, Rīga Stradiņš University, 1067 Riga, Latvia; zaiga.nora@rsu.lv (Z.N.-K.); anda.vilmane@rsu.lv (A.V.); 6Helsinki University Hospital, 00280 Helsinki, Finland; kim.vettenranta@helsinki.fi; 7College of Medicine, University of Malawi, Blantyre 3, Malawi; cmangani@medcol.mw

**Keywords:** parvovirus, bufavirus, tusavirus, cutavirus, gastroenteritis, respiratory-tract infection, leukemia, PCR, serology

## Abstract

Three human protoparvoviruses, bufavirus (BuV), tusavirus (TuV) and cutavirus (CuV), have recently been discovered in diarrheal stool. BuV has been associated with diarrhea and CuV with cutaneous T-cell lymphoma, but there are hardly any data for TuV or CuV in stool or respiratory samples. Hence, using qPCR and IgG enzyme immunoassays, we analyzed 1072 stool, 316 respiratory and 445 serum or plasma samples from 1098 patients with and without gastroenteritis (GE) or respiratory-tract infections (RTI) from Finland, Latvia and Malawi. The overall CuV-DNA prevalences in stool samples ranged between 0–6.1% among our six patient cohorts. In Finland, CuV DNA was significantly more prevalent in GE patients above rather than below 60 years of age (5.1% vs 0.2%). CuV DNA was more prevalent in stools among Latvian and Malawian children compared with Finnish children. In 10/11 CuV DNA-positive adults and 4/6 CuV DNA-positive children with GE, no known causal pathogens were detected. Interestingly, for the first time, CuV DNA was observed in two nasopharyngeal aspirates from children with RTI and the rare TuV in diarrheal stools of two adults. Our results provide new insights on the occurrence of human protoparvoviruses in GE and RTI in different countries.

## 1. Introduction

*Parvoviridae* is a family of small nonenveloped single-stranded DNA viruses that infect host-specifically many diverse animal species. There are two known human pathogens: parvovirus B19 (B19V), causing erythema infectiosum, arthritis, anemias and fetal death; and human bocavirus (HBoV) 1, causing pediatric respiratory-tract infections (RTI) and, infrequently, encephalitis. Furthermore, parvovirus 4 and HBoV2-4 infect humans with unclear disease associations [1]. In 2012–2016, metagenomic studies revealed, in human diarrheal stool samples, three more parvoviruses belonging to the *Protoparvovirus* genus: bufavirus (BuV), tusavirus (TuV) and cutavirus (CuV) [2,3,4]. Currently there are three known genotypes of BuV [5,6]. However, the clinical impact of the newly discovered protoparvoviruses is largely unknown. 

BuV DNA has worldwide been detected by PCR in low prevalence (0–4%) in diarrheal stool samples, whereas non-diarrheal stools have mainly been BuV-DNA negative [5,6,7,8,9,10,11,12,13,14,15,16,17,18,19]. However, the causative role of BuV in gastroenteritis (GE) remains unclear. Currently, TuV DNA has only been reported in the stool of a single child with unexplained diarrhea from Tunisia [3]. No other studies of TuV DNA have to our knowledge been published and further investigation of TuV in human samples is thus warranted. CuV is the newest parvovirus discovered in humans. Its prevalence in stool has hitherto not been reported since the original discovery, which conveyed a low prevalence (1–2%) in Brazilian and Botswanan diarrheic children [4]. Excitingly, dermal CuV DNA is documented to be associated with cutaneous T-cell lymphoma (CTCL) [20,21,22]. It has also been detected in skin biopsies of melanoma and organ transplant patients, but not in those of healthy subjects [21,23,24]. As most CuV studies have focused on screening malignant skin tissues, the prevalence of virus in stool and respiratory samples from respective GE and RTI patients has, unlike for BuV, not been investigated since its discovery, so nothing is known of acute CuV infections. 

The aim of the current study was to elucidate how common acute human protoparvovirus infections are in pediatric and adult patients with and without GE or RTI, to identify age- or gender-related and geographic distributions, and to find potential disease associations to GE or RTI. We analyzed in total 1072 stool samples and 316 respiratory samples from patients from six cohorts in three countries, Finland, Latvia and Malawi, for CuV, TuV and BuV DNA, as well as 445 plasma or serum samples for the respective seroprevalences in children from Latvia and Finland. For all children with protoparvovirus IgG-positive serum or plasma, we analyzed also their specific IgG by competition enzyme immunoassay (EIA) [25].

## 2. Materials and Methods

### 2.1. Study Cohorts

The Helsinki-B cohort from Finland comprises stool samples originally sent for bacterial diagnosis from 212 patients (age 1–94 years, median 36 years) with GE (Table 1). The samples had been analyzed during October 2012–March 2013 mainly for *Salmonella* spp., *Shigella* spp., *Campylobacter* spp., *Yersinia* spp., *Vibrio cholerae* and pathogenic *Escherichia coli* subtypes by culture or PCR. A bacterial pathogen was detected in 73/212 patient samples.

The Helsinki-V cohort from Finland comprises stool samples originally sent for viral diagnosis from 285 patients (age 0–99 years, median 74 years) with GE (Table 1). The samples had been tested for either norovirus alone, or for norovirus, adenovirus, rotavirus and astrovirus, during April–June 2013, by RT-PCR or antigen detection assay (Diarlex MB, Orion Diagnostica, Espoo, Finland). A viral pathogen was discovered in 99/285 samples. All the Helsinki-B and Helsinki-V samples had been sent for routine testing to the Helsinki University Hospital Laboratory (HUSLAB) from diverse locations in Finland, and thus were not from a few isolated outbreaks. Further, all the samples from these two cohorts have been previously studied for BuV [12]. The Ethics Committee of the Hospital District of Helsinki and Uusimaa approved the study of the deidentified samples.

The Tampere cohort includes 228 stool, nasal swab, and serum samples from 228 children (age 0–15 years, median 1.3 years) with GE (*n* = 42), RTI (*n* = 104), or both (*n* = 82) (Table 1). The patients were recruited between September 2009 and August 2011 in Tampere, Finland, and the samples were collected throughout the year [26]. Further, all the stool and swab samples from this cohort have been previously studied for BuV DNA and the sera for BuV and TuV IgG [13]. A written informed consent was obtained from the parents of all the children enrolled. The study was approved by the Ethics Committee of Pirkanmaa Hospital District, and it was conducted in accordance with the relevant guidelines and regulations.

The Latvia cohort consists of 44 nasopharyngeal aspirate (NPA), 115 stool and 102 plasma samples from 159 children (age 0–5 years, median 1.75 years), with GE (*n* = 62), RTI (*n* = 80), or both (*n* = 17) (Table 1). The study protocol was approved by the Ethics Committee of the Rīga Stradiņš University [27]. Written informed consent was received from all parents or guardians of the participating children.

The Malawi cohort includes 168 stool samples collected from 164 healthy or diseased children (age 6–12 months) from rural Malawi for a Child Nutrition Intervention Study (LCNI-5) between January 2008 and November 2009 (Table 1). The study adhered to the principles of the Declaration of Helsinki and regulatory guidelines in Malawi. Written informed consent was obtained from the participants’ guardians and the trial protocol was approved by the research ethics committees of the University of Malawi College of Medicine and of the Pirkanmaa Hospital District, Finland.

The leukemia cohort comprises 44 nasal swab, 115 serum and 64 stool samples from 50 children (age 0.4–15.3 years, median 5.7 y) with acute leukemia undergoing anticancer treatment, collected between April 2000 and October 2005 at 4 Finnish university hospitals (Table 1) [28]. The study was approved by the Ethical Committees of the Medical faculties of Turku, Oulu, Kuopio, and Helsinki Universities. Written informed consent was obtained from the patients or from their parents.

### 2.2. CuV-BuV-TuV qPCR Assay

To detect and quantify CuV, BuV, and TuV DNA, a multiplex real-time qPCR was performed with primers and hydrolysis probes located in the VP2 regions of CuV and TuV and the NS1 region of BuV, as previously described [21]. The samples from the Helsinki and Tampere cohorts [12,13], with previously published BuV qPCR results, were analyzed by a duplex qPCR assay for CuV and TuV DNA. For most of the samples, an initial screening was done in multiplex assays and positives were repeated in singleplex. For two CuV DNA-positive samples (from the Helsinki-V and Tampere cohorts each), the qPCR assay could, however, not be repeated due to insufficient amounts of sample. Further, sufficient volumes were not available for 49 Helsinki-V samples, including a TuV DNA-positive one, for the initial qPCR screening and water was added to make up to 5 µl for PCR. All positive samples were confirmed by cloning and sequencing (Appendix A). 

### 2.3. CuV-BuV1-3-TuV IgG EIA

A total of 115 serum and 102 plasma samples from 29 leukemic children and 102 Latvian children, respectively, were screened for CuV-BuV1-3-TuV IgG, and the 228 serum samples from the Tampere children were screened for BuV2 and CuV. BuV and TuV IgG EIA results for these Tampere children have been previously published, but due to frequent cross-reactivity between BuV2 and CuV IgG, learned since then, these samples were here screened besides for CuV, also for BuV2 IgG, with cross blocking [13]. The samples were analyzed by an in-house IgG EIA, with biotinylated VP2 virus-like particles (VLP) as antigen, and all samples with an OD > 0.1 were confirmed using a competition assay as previously described [25], but with a 40-min incubation at room temperature with substrate 3,3′,5,5′-tetramethylbenzidine (BD OptEIA™, Franklin Lakes, NJ, USA). Optical densities (ODs) were measured at 450 nm (Multiskan EX; Thermo Fischer Scientific, Pittsburgh, PA, USA). Samples with OD between 0.1–0.24 were considered to be borderline positives. 

### 2.4. Statistical Analysis

Statistics were calculated with IBM SPSS Statistics v25 (IBM Corp, NY, USA) for Pearson’s χ2. A *p* value <0.05 was considered statistically significant.

## 3. Results

### 3.1. CuV DNA in Stool and Respiratory Samples

In all, CuV DNA was detected in stool samples from 26/1039 (2.5%) adults and children in the current study (Table 2). The prevalence of CuV DNA in stool samples among the six cohorts was between 0–6.1%. The viral loads of CuV DNA-positive stool samples varied between 1.24 × 10^2^ and 1.05 × 10^4^ copies/mL stool supernatant (Table 3). No statistical differences in the CuV viral loads were observed between the different cohorts and between patients with GE or RTI.

Among the observed country-specific CuV-DNA prevalences in stool samples of children <15 years, Latvia had the highest prevalence (7/115, 6.1%) followed by Malawi (7/164, 4.3%) and Finland (1/341, 0.3%) (Figure 1A). The CuV-DNA prevalence in Latvian and Malawian children was significantly higher than in the Finnish children (*p* < 0.05). However, there was no significant differences in CuV DNA prevalence between children in different age groups of all the three cohorts; 0–1 years (10/291, 3.4%), >1–5 years (4/192, 2.1%), >5–15 years (1/24, 4.2%) (*p* ≥ 0.5). Nevertheless, when considering both adults and children from all cohorts in Finland (*n*= 761) (including Helsinki-B, Helsinki-V, Tampere and Leukemia cohorts), the CuV DNA prevalence among individuals above 60 years of age (5.1%, 11/214) was significantly higher (*p* < 0.001) than that of individuals below 60 years of age (0.2%, 1/546) (Figure 1B). This was also significant when including adults and children from only the 2 Helsinki cohorts. 

Among children (<15 years) from the Helsinki-B, Helsinki-V, Tampere and Latvia cohorts, CuV DNA was detected in stools of 4/180 (2.2%) with GE alone, of 2/183 (1.1%) with RTI alone, of 2/99 (2%) with symptoms of both. The difference in CuV-DNA prevalence between the GE and RTI groups was not significant (*p* > 0.5). Further, stool samples from 4 healthy children from Malawi were CuV-DNA positive (Table 3). 

Among the respiratory samples, CuV DNA was detected in 2/44 (4.5%) NPAs from children from Latvia but not in any of 272 pediatric nasal swabs from the Leukemia or Tampere cohorts from Finland. The two CuV DNA-positive NPAs from the Latvian cohort were from children 45- and 39-months of age, both with RTI and fever at the time of sampling. The viral loads of the CuV DNA-positive NPA samples were 8.62 ×10^2^ and 2.04 × 10^3^ copies/mL of NPA respectively (Table 3). No stool samples were available from these two individuals and the corresponding plasma samples were CuV-DNA and -IgG negative. In addition, they were both positive for HBoV1 DNA in NPA, but not in serum, and one of the children was also positive for human rhinovirus RNA. 

The 91-nt long CuV amplicons from all CuV DNA-positive samples in the current study, had 1 to 5 nucleotide mismatches in the non-primer-binding region, compared with the CuV sequence NC_039050.1, serving as positive control in qPCR (Appendix A).

### 3.2. BuV and TuV DNA in Stool and Respiratory Samples

BuV DNA in stools was analyzed only from the Latvia, Malawi and Leukemia cohorts, where 1/314 (0.3%) children were PCR positive. The single BuV DNA-positive stool sample had a virus load of 1.23 × 10^3^ copies/mL and was from a healthy child, 6 months of age, from Malawi (Table 3). Of note, we did not analyze for BuV DNA in the samples from the Helsinki and Tampere cohorts because they had already been analyzed in our previous study, where it was detected in stools of the Helsinki-V (3/386, 0.7%), Helsinki-B (4/243, 1.6%), and Tampere cohorts (3/955, 0.3%), and in a nasal swab (1/955, 0.1%) of a child from Tampere [12,13]. Furthermore, none of the current respiratory samples from 76 children in the Latvia or Leukemia cohorts contained BuV DNA (Table 2).

TuV DNA was detected in stool samples from 2/1039 (0.2%) individuals but in none of the 316 respiratory samples (Table 2). The TuV DNA-positive stool samples were from 22- and 27-year-old women in the Helsinki-V and -B cohorts, respectively, from Finland. The viral loads of the samples were 8.90 × 10^1^ and 4.42 × 10^1^ copies/mL stool suspension, respectively (Table 3). The young adults had gastrointestinal symptoms, but no other causative pathogens were found. Further, the 27-year-old woman had recently travelled to Jamaica and Turkey (Istanbul). In the sequence alignment, the 118-nt long TuV amplicons had 5 and 6 nucleotide mismatches in the non-primer-binding target region, respectively, compared to the only known TuV sequence KJ495710.1, also serving as positive control in qPCR (Figure 2). There were no more stool materials left for further sequencing of longer amplicons. 

### 3.3. Protoparvovirus IgG in Children

We screened 228 serum samples from the Tampere children for CuV IgG and the seroprevalence was 2/228 (0.9%). For the two CuV IgG-positive samples, the corresponding nasal swab and stool samples from the same patients were CuV-DNA negative (Table 4). Further, from one child whose stool was CuV DNA-positive, we could not detect CuV DNA in the swab or CuV DNA or IgG in serum. For the 102 Latvian children, the CuV seroprevalence was 2.9% (3/102). All three CuV-seropositive children (La00, La78 and La34) had GE symptoms, but we did not detect CuV DNA in their stool samples, and NPA samples were not available from these three children. Further, three other children (La82, La03 and La14) with CuV DNA-positive stools and two children (La49 and La50) with CuV DNA-positive NPA samples were seronegative. No follow-up samples were available. 

No BuV1, 2, 3, or TuV IgG antibodies were detected in 102 plasma samples from the Latvian children. BuV1, 2, 3 and TuV IgG EIA results for 228 serum samples from the Tampere children have been previously published and seven children (7/228, 3.1%) had BuV IgG and one child (1/228, 0.4%) had TuV IgG [13]. Of the BuV IgG-positive cases, one had BuV1, three had BuV2, two had BuV3 and one had both BuV1 and 2. However, due to the known cross-reactivity of BuV2 and CuV IgG, we re-analyzed the four previously reported BuV2 IgG-positive samples by competition EIAs and proved two of them to actually be CuV IgG and not BuV2 IgG, whereas the 2 remaining positives were truly BuV2 IgG. Therefore, the correct BuV IgG seroprevalence among the Tampere children is 2.6% (6/228).

A total of 115 serum samples from 29 leukemic children were screened for CuV, BuV1-3, and TuV IgG. One child (1/29, 3.4%) had BuV2 IgG and four children (4/29, 13.8%) had slightly raised CuV IgG absorbances. Multiple serum samples were available from three CuV IgG-positive children and the observed OD values remained similar for the whole follow-up period, up to one month for 2 children and one year for one child. Overall, the CuV absorbance values were low (0.14–0.27), but specific by competition EIA, and the corresponding stool and swab samples from these children were CuV-DNA negative. No BuV1, BuV3, or TuV IgG antibodies were detected in this cohort.

## 4. Discussion

CuV has already gained much interest due to its association with cutaneous T-cell lymphoma, despite being discovered only in 2016. It has been predominantly searched for in skin cancers [4,20,21,22], however, nothing is known of its acute infections. No respiratory or stool samples have been analyzed for its prevalence or to characterize its possible symptoms in acute infections. Likewise, with TuV, no PCR results have been reported in human samples apart from the initial paper where it was discovered in the stool of a single Tunisian child with unexplained diarrhea [3]. Hence in the current study, we analyzed CuV and TuV DNA prevalences and their age- or gender-related and geographic distributions in human stool and respiratory samples of patients, with and without GE or RTI, and compared the findings with those of the more studied BuV. 

The highest CuV DNA prevalence in stool samples was detected among the Latvian children as compared to all other cohorts analyzed in both this and the original discovery study [4]. The observed CuV viral loads in stool samples, in this study, were similar to those of BuV from previous studies of stool samples from GE and RTI patients, including both adults and children [12,13]. Low viral loads in stool samples of these protoparvoviruses may be remnants from a previous infection. Nevertheless, no known pathogens were detected in the majority of the CuV DNA-positive adults and children with GE, allowing speculations for a possible causal role of CuV. However, further evidence from serological studies together with viral DNA detection in paired samples is required to provide additional clues about CuV acute infections in humans and its possible role in GE.

CuV was significantly more prevalent in stool samples from Finnish GE patients above 60 years than below. By contrast, the BuV-DNA stool-positive samples were from a broader age group of Finnish adults, of 21–89 years, with a median of 53 years [12]. Unfortunately, stool samples from adults from Latvia and Malawi were not available for similar prevalence comparisons in these two countries. Further, we did not have serum samples from the adults for CuV IgG prevalence comparisons in different age groups.

Interestingly, we observed for the first time CuV DNA in NPAs of children with RTI, indicating that CuV is shedding into the airways. However, due to the low prevalence, we did not observe a significant difference (*p* > 0.5) in the CuV airway genoprevalences between the GE and RTI groups, which in our association studies served as controls for each other. Due to the lack of association, the low DNA prevalence and copy numbers, and the presence of other RTI-causing pathogens, we did not see a causal role for CuV in RTI. 

CuV has been more widely studied in skin samples than in stool or respiratory samples. So far, CuV DNA has been detected in skin biopsies of CTCL, melanoma and transplant patients with prevalences between 1.1–16% [4,21,22,23], but not in those of healthy subjects [4,21]. However, viral DNA has surprisingly been found in skin surface swabs of healthy adults (9/237, 3.8%) and of HIV-positive men (35/205, 17.1%) [24]. Recently, CuV DNA was found in the skin biopsy of an acute lymphoblastic leukemia patient [29]. The overall occurrence of CuV DNA in skin seems to be more common than in stool or respiratory samples, which may be because the virus persists in skin tissues for years [21], whereas shedding may be limited to the acute phase of infection. 

Previously, the CuV-IgG prevalence was observed to be low, ranging from 0–6% among healthy individuals in the USA, Iran, Iraq and Finland, with the exception of 9.5% in CTCL patients [21,25]. The observed CuV seroprevalences in the Tampere (0.9%) and Latvian (2.9%) children are thus in line with previous studies. CuV DNA was detected relatively often in skin and stool when taking into consideration the observed lower seroprevalence in any population worldwide. This could be interpreted in three ways: (i) B-cell immunity is not induced due to the infections being local; (ii) some antibody responses wane with time below the detection limit, as seen for human bocavirus [30]; or (iii) a prior BuV infection could inhibit the induction of CuV IgG due to an immunological phenomenon called original antigenic sin or imprinting, also seen among the related human bocaviruses [31]. However, the first scenario seems logical only for stool positivity, since CuV persists in malignant skin tissue and lymph nodes [21], the middle scenario seems the most logical unless the whole virion persists or viral VP2 is expressed, which would rather boost immunity, and the latter would be more expected in Asia and Africa, where the BuV seroprevalence is much higher [25]. In contrast to our previous skin study, we observed no correlation between CuV DNA positivity in stool or respiratory samples and CuV IgG positivity in the same individuals [21]. This could be due to scenario (i) above, or more likely, as for other human parvoviruses, to acute systemic infections occurring before the antibody responses become measurable. Unfortunately, we did not have follow-up sera to observe possible seroconversions.

In our earlier study, we revealed a high BuV seroprevalence in Kenyan adults of 72% (compared to 2% in Finland) and in children 21% [25], nevertheless, in our current study, we found only one child from Malawi with BuV DNA in stool. This could be due to the lower age of the children, or to regional differences within Africa. BuV has been associated with GE, as most BuV DNA-positive cases have been diarrheic [6]. However, no such symptoms had been described for the single BuV DNA-positive child in Malawi. The BuV-DNA prevalence in stool was lower than that of CuV in the Malawian children and no BuV DNA was detected in the Latvian children, who had the highest CuV-DNA prevalence. This would suggest that the prevalence of BuV in these countries is much lower than that of CuV. The low BuV-DNA prevalence in children is supported by previous studies where all or a majority of the BuV DNA-positive stool samples were from adults [6,9,10,12,15].

This is the first published study of TuV DNA in human stool and respiratory samples since the initial metagenomics study [3]. TuV DNA was detected, though in low quantities, in stool samples from two adults from Finland with GE. There is currently only one TuV sequence published, on which our qPCR and VP2-VLP EIAs were based on [3]. Further, the observed 5 and 6 nucleotide mutations in the non-primer binding region of the 118-nt-long amplicon sequences from the two TuV DNA-positive stools, indicate that TuV may be more diverse in nature than previously believed. The low prevalence could therefore be attributed to non-optimal primers or probe that affects the sensitivity of the current PCR method. This could also explain the low TuV geno- and seroprevalence in the populations studied so far [13,25]. Unfortunately, there was not enough sample material to sequence more of the genome. More studies are therefore needed to confirm whether there are different strains of TuV circulating in the population and its true prevalence globally. 

Unlike for TuV, there are many published sequences for CuV, which differ only by a few nucleotides in the region covering our primer and probe sites [4,21]. Hence with a degenerate CuV forward primer and probe in our qPCR assays, we were able to detect also variant strains of CuV. However, apart from the known mismatches, there could be CuV strains with other mutations at the primer and probe binding sites, resulting in lack of detection with our current qPCR assay. 

Children with acute leukemia are more prone to opportunistic infections due to cytotoxic and immunosuppressive treatments [32]. Hence, in our study, we wanted to see the geno- and seroprevalences of protoparvoviruses in leukemic children. However, none of the 64 stool samples and 44 swab samples from 35 and 32 leukemic children, respectively, were CuV, TuV or BuV DNA positive in our current study. Interestingly, the CuV IgG seroprevalence of leukemic children, despite their immunosuppressive state, was 13.8% in our study, one of the highest observed in any cohort so far [21,25]. By competition EIA, we observed that the CuV-specific reactivity was blocked completely only by homotypic CuV antigen and not by heterotypic antigen (BuV1 and 2), thereby confirming specificity. Despite the confirmatory blocking results, the overall absorbance values were unusually low and therefore the results should be interpreted with caution. As these children with acute leukemia had not received blood transfusions, the observed higher CuV IgG prevalence and overall lower absorbance values could be attributed to their immunosuppressive state.

## 5. Conclusions

Since their discoveries, we observed for the first time CuV DNA in nasal swabs and report the genoprevalences of CuV and TuV in stool and respiratory samples. CuV DNA was detected relatively often when taking into consideration the observed seroprevalence. The overall low geno- and seroprevalences and low DNA loads, together with a lack of association to GE or RTI, indicate that CuV is not at least a frequent cause of GE or RTI. However, the reason for viral shedding in the airways and stool in different populations and the ability of these newly discovered human protoparvoviruses to cause diseases remain to be elucidated.

## Figures and Tables

**Figure 1 viruses-13-00483-f001:**
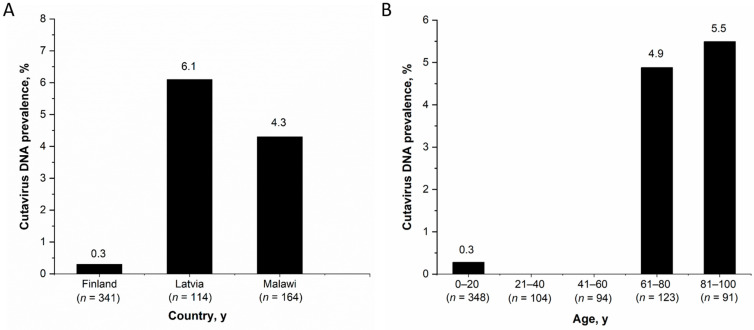
CuV-DNA stool prevalence (**A**) Among children from Finland, Latvia and Malawi, (**B**) Among all patients from Finland by age.

**Figure 2 viruses-13-00483-f002:**
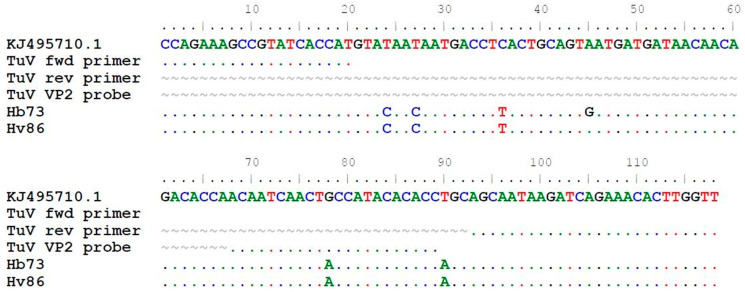
Alignment of the 118 nt sequenced qPCR amplicons from TuV DNA-positive stool samples in the current study to the only published TuV reference sequence KJ495710.1 (3085–3202 bp). Dot (.) indicates identical nucleotides and tilde (~) lacking nucleotides.

**Table 1 viruses-13-00483-t001:** Characteristics of cohorts used in the current study.

Cohort	No. of Individuals Studied (N = 1098)	Health Status *	Age Range (Median age, y)	Gender	Time of Sample Collection	Sample Type Analyzed *
Helsinki-B	212	GE	1 to 94 years (36)	Female, N = 113, Male, N = 99	October 2012–March 2013	Stool
Helsinki-V	285	GE	0 to 99 years (74)	Female N = 154, Male N = 131	April-June, 2013	Stool
Tampere	228	GE and/or RTI	0 to 15 years (1.3)	Female, N = 94, Male, N = 134	September 2009–August 2011	Stool, nasal swab and serum
Latvia	159	GE and/or RTI	0 to 5 years (1.7)	Female, N = 59, Male, N = 100	November 2013–April 2017	Stool, NPA and plasma
Malawi	164	Healthy or diseased	6 or 12 months	Female, N = 73 Male, N = 91	January 2008–November 2009	Stool
Leukemia	50	AML or ALL	0 to 15 years (5.7) ^†^	Female, N = 17 Male, N = 33	April 2001–October 2004	Stool, nasal swab and serum

* GE, gastroenteritis; ALL, acute lymphocytic leukemia; AML, acute myeloid leukemia; RTI, respiratory tract infection; NPA, nasopharyngeal aspirate. ^†^ Age not available for 4 children from this cohort and hence are not included in the median age calculation.

**Table 2 viruses-13-00483-t002:** Prevalence of CuV, BuV and TuV DNA in stool and respiratory samples in different population cohorts.

Cohort	Sample Type	CuV DNA Positive Patients (Prevalence; 95% CI)	BuV DNA Positive Patients (Prevalence; 95% CI)	TuV DNA Positive Patients (Prevalence; 95% CI)
Helsinki-B	Stool	1/212 (0.5%; 0.0–2.7)	4/243 (1.6%; 0.0–4.5) ^†^	1/212 (0.47%; 0.0–2.7)
Helsinki-V ^¶^	Stool	10/285 (3.5%; 0.6–7.3)	3/386 (0.7%; 0.0–2.0) ^‡^	1/285 (0.35%; 0.0–1.7)
Tampere	Stool	1/228 (0.4%; 0.0–2.4)	3/955 (0.3%; 0.0–0.5) ^§^	0/228
	Nasal swab	0/228	1/955 (0.1%; 0.0–0.2) ^§^	0/228
Latvia	Stool	7/115 (6.1%; 2.2–15.1)	0/115	0/115
	NPA	2/44 (4.5%; 0.0–20.1)	0/44	0/44
Malawi	Stool	7/164 (4.3%; 0.8–10.4)	1/164 (0.6%; 0.0–3.7)	0/164
Leukemia	Stool	0/35	0/35	0/35
	Nasal swab	0/32	0/32	0/32

^†^ 243 samples from this cohort were previously studied for BuV DNA (4/243, 1.6%) [12]. Due to unavailability, 31 samples could not be analyzed for CuV and TuV DNA in the current study. ^‡^ In addition to 285 samples analyzed in the current study, 101 samples from this cohort were previously analyzed for BuV DNA (3/386, 0.7%) [12]. Due to unavailability, these 101 samples could not be analyzed for CuV and TuV DNA in the current study. ^§^ In addition to 228 samples analyzed in the current study, 727 samples from this cohort were previously analyzed for BuV DNA [13]. So overall BuV DNA prevalence was 1/955 (0.1%) in nasal swab and in 3/955 (0.3%) stools. ^¶^ Water was added to make up to 5 μL volume for PCR reaction in 49 samples from this cohort due to lower sample volume available. None of these 49 samples were CuV DNA-positive, but one sample was TuV-DNA positive. The prevalences of CuV and TuV excluding these 49 samples were 4.2% and 0% respectively. CuV, cutavirus; BuV, bufavirus; TuV, tusavirus; NPA, nasopharyngeal aspirate.

**Table 3 viruses-13-00483-t003:** CuV, BuV and TuV DNA-positive samples from children and adults in the current study.

Patient No.	Copies per ml ^†^	Age/Gender	Sampling Date	Other Pathogens Found	Symptoms and Signs
CuV DNA positives					
Hb40	1.69 × 10^2^	79 years/F	2013 Feb 07		GE
Hv61	7.99 × 10^3^	77 years/F	2013 Apr 18	-	GE
Hv99	1.05 × 10^4^	89 years/M	2013 Apr 16	-	GE
Hv92	NA	90 years/M	2013 Apr 20	NoV	GE
Hv57	2.65 × 10^2^	78 years/M	2013 Apr 29	-	GE
Hv93	3.57 × 10^2^	87 years/M	2013 May 04	-	GE
Hv05	1.93 × 10^2^	80 years/M	2013 May 02	-	GE
Hv65	9.61 × 10^2^	77 years/M	2013 May 10	-	GE
Hv69	1.30 × 10^3^	81 years/M	2013 May 13	-	GE
Hv78	9.93 × 10^3^	67 years/M	2013 May 13	-	GE
Hv31	1.24 × 10^2^	99 years/F	2013 May 24	-	GE
Ta13	1.43 × 10^3^	10 years/M	2009 May 04	-	GE/RTI
La42	1.50 × 10^2^	1 year/F	2015 Jan 20	RSVB	RTI, Fever
La46	3.22 × 10^3^	1 year/M	2015 Feb 03	HBoV1	RTI, Fever
La82	9.28 × 10^2^	7 months/M	2016 Jan 26	-	GE, fever, RTI
La03	5.15 × 10^2^	3 years/M	2016 Mar 24	HBoV1 and RoV	GE
La14	1.40 × 10^3^	2 years/M	2016 May 19	-	GE and fever
La43	1.37 × 10^3^	2 years/M	2016 Oct 05	-	GE and fever
La63	6.27 × 10^3^	3 years/M	2016 Nov 03	NoV	GE and fever
La49	8.62 × 10^2 ‡^	45 months/M	2016 Oct 18	HBoV1	Fever, RTI
La50	2.04 × 10^3 ‡^	39 months/M	2016 Oct 19	HBoV1; HRV	Fever, RTI
Ma25	9.57 × 10^2^	6 months/M	2008 Apr 07	-	Data not available
Ma40 ^§^	4.35 × 10^2^	1 year/M	2008 Dec 18	-	Healthy
Ma61	5.44 × 10^3^	1 year/F	2008 Dec 31	-	Data not available
Ma99	1.41 × 10^3^	1 year/M	2009 Jan 14	-	Healthy
Ma66	2.20 × 10^3^	1 year/M	2009 Mar 17	-	Healthy
Ma45	2.76 × 10^3^	6 months/F	2008 Nov 27	-	Fever
Ma06	5.21 × 10^2^	6 months/F	2009 Jan 16	-	Healthy
BuV DNA positives^#^					
Ma14	1.23 × 10^3^	6 months/M	2009 Jul 02	-	Healthy
TuV DNA positives					
Hb73 ^¶^	4.42 × 10^1^	27 years/F	2013 Feb 25	-	GE
Hv86	8.90 × 10^1^ **	22 years/F	2013 May 14	-	GE

^†^ Copies expressed as copies per ml of stool supernatant or nasopharyngeal aspirate (NPA). ^‡^ Nasopharyngeal aspirate (NPA). ^§^ A stool sample taken 6 months before December 18, 2008 from this child was CuV DNA negative. ^¶^ Hb73 travelled to Jamaica and Turkey (Istanbul). ^#^ Only one BuV-positive sample was found in our current study, because the Helsinki and Tampere cohorts have been studied for BuV DNA before [12,13]. ** Water was added to the sample up to 5 µL for qPCR reaction. Hence actual quantity may vary. NA, not available; GE, gastroenteritis; RTI, respiratory tract infection; RSVB, respiratory syncytial virus B; HBoV1, human bocavirus 1, HRV, human rhinovirus; NoV, norovirus; RoV, rotavirus.

**Table 4 viruses-13-00483-t004:** Comparison of CuV DNA in stool and respiratory samples with CuV IgG from CuV DNA- or IgG-positive individuals.

Patient (Symptoms)	Sample Type	Sampling Date of Stool/NPA and sera	CuV DNA (Copies/mL of Supernatant)	CuV IgG EIA Absorbances ^‡^
La49 (RTI)	Stool		NA	0.038
	NPA	2016 Oct 18	Pos (8.62 ×10^2^)	
La50 (RTI)	Stool		NA	0.039
	NPA	2016 Oct 19	Pos (2.04 ×10^3^)	
La42 (RTI)	Stool	2015 Jan 20	Pos (1.50 × 10^2^)	NA
	NPA	2015 Jan 20	Neg	
La46 (RTI)	Stool	2015 Feb 03	Pos (3.22 × 10^3^)	NA
	NPA		NA	
La82 (GE and RTI)	Stool	2016 Jan 26	Pos (9.28 × 10^2^)	0.018
	NPA		NA	
La03 (GE)	Stool	2016 Mar 24	Pos (5.15 × 10^2^)	0.015
	NPA		NA	
La14 (GE)	Stool	2016 May 19	Pos (1.40 × 10^3^)	0.527 ^†^
	NPA		NA	
La43 (GE)	Stool	2016 Oct 05	Pos (1.37 × 10^3^)	NA
	NPA		NA	
La63 (GE)	Stool	2016 Nov 03	Pos (6.27 × 10^3^)	NA
	NPA		NA	
La00 (GE)	Stool	2016 Mar 04	Neg	0.228
	NPA		NA	
La78 (GE)	Stool	2016 Jan 15	Neg	0.976
	NPA		NA	
La34 (RTI)	Stool		NA	0.326
	NPA			
Ta13 (GE and RTI)	Stool	2009 Nov 04	Pos (1.43 × 10^3^)	0.050
	Nasal swab	2009 Nov 04	Neg	
Ta47 (RTI)	Stool	2009 Sep 28	Neg	0.873
	Nasal swab	2009 Sep 28	Neg	
Ta69 (RTI)	Stool	2011 Feb 21	Neg	2.683
	Nasal swab	2011 Feb 20	Neg	

^†^ Although the absorbance value observed was 0.527, the absorbance could not be blocked with homotypic or heterotypic VLPs in competition EIA, and hence this sample was considered CuV IgG negative. ^‡^ All the serum and plasma samples were collected on the same date as their respective stool or NPA or NS samples. EIA, enzyme immunoassay; GE, gastroenteritis; RTI, respiratory tract infection; NA, not available; NPA, nasopharyngeal aspirate; NS, nasal swab.

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
