# Peer review of "Human Protoparvovirus DNA and IgG in Children and Adults with and without Respiratory or Gastrointestinal Infections"

_viruses, 2021, doi:10.3390/v13030483_

Round 1

Reviewer 1 Report

The study describes the serological and molecular screening of cutavirus and tusavirus in human patients  with gastroenteric (GE) and respiratory tract (RT) symptoms from different sample collections from different geographical latitudes. The manuscript is well-written and provides interesting information of circulation of human protoparvoviruses in humans without firmly confirming the association with GE or RT disease.  I suggest to publish the manuscript unless the authors provide major details on ethical approvals of studies from different collection of samples.

Author Response

Dear Editors,

Thank you for this possibility for us to correct our manuscript. We have now added the “(EIA)” abbreviation to enzyme immunoassay, and answered Reviewers questions below (A). We have further re-written some minor mistakes in our text and expressed our wish to move Table 1 not to be divided onto two pages. Further we fixed the alignment of Table 4. All changes are shown with Track changes.

Reviewer 1

The study describes the serological and molecular screening of cutavirus and tusavirus in human patients  with gastroenteric (GE) and respiratory tract (RT) symptoms from different sample collections from different geographical latitudes. The manuscript is well-written and provides interesting information of circulation of human protoparvoviruses in humans without firmly confirming the association with GE or RT disease.  I suggest to publish the manuscript unless the authors provide major details on ethical approvals of studies from different collection of samples.

Dear Reviewer 1,

A: Thank you very much for the comments.

Sincerely,

Ushanandini Mohanraj

Reviewer 2 Report

In the present study, Mohanraj and colleagues detected the DNA prevalence of three human protoparvoviruses, bufavirus (BuV), tusavirus (TuV) and cutavirus (CuV), in stool and nasal swabs from 1098 patients with or without GE or RTI symptoms from Finland, Latvia, and Malawi. Some serum or plasma samples were also detected for viral IgG. They showed that the overall CuV DNA prevalence in stool samples is between 0-6.1%. They found that CuV DNA is more prevalent is patients older than 60 years in Finland. CuV DNA was also observed more prevalent in stool samples in Latvian and Malawian children than Finnish children. The manuscript is well written and provide important information of the prevalence of human protoparvoviruses include CuV, TuV, and BuV in GE and RTI patients.

I only have some minor comments to the authors.

  1. What’s the possible reason why CuVs were mainly detected in children and people older than 60 years old (Figure 1b)?
  2. Line 141, introduce EIA when it first appear in the manuscript.
  3. Line 258-264, two of the four BuV2 IgG were CuV IgG, which makes the correct BuV IgG positive children are 5, not 6. Please double check the numbers.
  4. Line 265-266, did the authors perform competition EIA for this one BuV2 IgG and four CuV IgG positive samples since there is cross-reactivity between BuV2 and CuV?

Author Response

Dear Editors,

Thank you for this possibility for us to correct our manuscript. We have now added the “(EIA)” abbreviation to enzyme immunoassay, and answered Reviewers questions below (A). We have further re-written some minor mistakes in our text and expressed our wish to move Table 1 not to be divided onto two pages. Further, we fixed the alignment of Table 4. All changes are shown with Track changes.

Reviewer 2

In the present study, Mohanraj and colleagues detected the DNA prevalence of three human protoparvoviruses, bufavirus (BuV), tusavirus (TuV) and cutavirus (CuV), in stool and nasal swabs from 1098 patients with or without GE or RTI symptoms from Finland, Latvia, and Malawi. Some serum or plasma samples were also detected for viral IgG. They showed that the overall CuV DNA prevalence in stool samples is between 0-6.1%. They found that CuV DNA is more prevalent is patients older than 60 years in Finland. CuV DNA was also observed more prevalent in stool samples in Latvian and Malawian children than Finnish children. The manuscript is well written and provide important information of the prevalence of human protoparvoviruses include CuV, TuV, and BuV in GE and RTI patients.

I only have some minor comments to the authors.

Dear Reviewer 2,

A: Thank you very much for your comments. We have addressed them as follows.

What’s the possible reason why CuVs were mainly detected in children and people older than 60 years old (Figure 1b)?

A: We do not know why in Finland, the CuV DNA prevalence was higher in over 60-year-olds than in children. It can be because its way of transmission favoring senior citizens, or because their weakened immune system. It may also be that the virus persists in tissues and then re-activates in the elderly, leading to excretion in stool.

Unfortunately, we didn’t have samples from adults from Latvia and Malawi where CuV prevalence was higher compared to Finland. We do not know if we would have observed a similar pattern as in Finland in these two other countries, so we dare not generalize.

Line 141, introduce EIA when it first appears in the manuscript.

A: This error has now been rectified. EIA is introduced in Line 78.

Line 258-264, two of the four BuV2 IgG were CuV IgG, which makes the correct BuV IgG positive children are 5, not 6. Please double check the numbers.

A: One child from our previous publication had both BuV1 and 2 IgG antibodies. After re-analysis with CuV IgG, the child had BuV1 and CuV IgG and not BuV2 as previously published. Hence after re-analysis, one had BuV1, two had BuV2, two had BuV3 and one had BuV1 and CuV. The correct number of BuV IgG positive children after re-analysis is 6.

Line 265-266, did the authors perform competition EIA for this one BuV2 IgG and four CuV IgG positive samples since there is cross-reactivity between BuV2 and CuV?

A: EIA cross blocking with CuV and BuV2 was performed for all BuV2 IgG- and CuV IgG-positive samples, respectively. No cross blocking was observed for these samples. Hence the observed BuV2- and CuV-IgG reactivities were specific. 

Sincerely,

Ushanandini Mohanraj